# Direct administration of the non-competitive interleukin-1 receptor antagonist rytvela transiently reduced intrauterine inflammation in an extremely preterm sheep model of chorioamnionitis

Yuki Takahashi[1,2]*, Masatoshi Saito[1,2], Haruo Usuda[1], Tsukasa Takahashi[1,2], Shimpei Watanabe[2], Takushi Hanita[2], Shinichi Sato[2], Yusaku Kumagai[2], Shota Koshinami[2], Hideyuki Ikeda[2], Sean Carter[1], Erin L. Fee[1], Lucy Furfaro[1], Sylvain Chemtob[3], Jeffrey Keelan[1], David Olson[4], Nobuo Yaegashi[2], John P. Newnham[1], Alan H. Jobe[1,5], Matthew W. Kemp[1,2,6,7]

1 Division of Obstetrics and Gynecology, The University of Western Australia, Perth, Western Australia, Australia, 2 Centre for Perinatal and Neonatal Medicine, Tohoku University Hospital, Sendai, Japan, 3 Department of Pharmacology and Physiology, University of Montreal, Montreal, Canada, 4 Department of Obstetrics and Gynaecology, University of Alberta, Alberta, Canada, 5 Cincinnati Children's Hospital Medical Centre, Cincinnati, OH, United States of America, 6 School of Veterinary and Life Sciences, Murdoch University, Perth, Western Australia, Australia, 7 Department of Obstetrics and Gynaecology, Yong Loo Lin School of Medicine, National University of Singapore, Singapore, Singapore

* yuki.sakurai@uwa.edu.au

## Abstract

### Background

Intraamniotic inflammation is associated with up to 40% of preterm births, most notably in deliveries occurring prior to 32 weeks' gestation. Despite this, there are few treatment options allowing the prevention of preterm birth and associated fetal injury. Recent studies have shown that the small, non-competitive allosteric interleukin (IL)-1 receptor inhibitor, rytvela, may be of use in resolving inflammation associated with preterm birth (PTB) and fetal injury. We aimed to use an extremely preterm sheep model of chorioamnionitis to investigate the anti-inflammatory efficacy of rytvela in response to established intra-amniotic (IA) lipopolysaccharide (LPS) exposure. We hypothesized that rytvela would reduce LPS-induced IA inflammation in amniotic fluid (AF) and fetal tissues.

### Methods

Sheep with a single fetus at 95 days gestation (estimated fetal weight 1.0 kg) had surgery to place fetal jugular and IA catheters. Animals were recovered for 48 hours before being randomized to either: **i)** IA administration of 2 ml saline 24 hours before 2 ml IA and 2 ml fetal intravenous (IV) administration of saline (Saline Group, n = 7); **ii)** IA administration of 10 mg LPS in 2 ml saline 24 hours before 2 ml IA and 2 ml fetal IV saline (LPS Group, n = 10); **3)** IA administration of 10 mg LPS in 2 ml saline 24 hours before 0.3 mg/fetal kg IA and 1 mg/fetal

**Data Availability Statement:** All relevant data are within the manuscript. Raw western images are in attached file.

**Funding:** DO, JK, MK. This work was supported by funding from the National Health and Medical Research Council (NHMRC; GNT1145295). MK. This work was supported by funding from the Women and Infants Research Foundation.

**Competing interests:** DO and SC are founders and directors of Maternica Therapeutics, which has a commercial interest in the development of rytvela.

kg fetal IV rytvela in 2 ml saline, respectively (LPS + rytvela Group, n = 7). Serial AF samples were collected for 120 h. Inflammatory responses were characterized by quantitative polymerase chain reaction (qPCR), histology, fluorescent immunohistochemistry, enzyme-linked inmmunosorbent assay (ELISA), fluorescent western blotting and blood chemistry analysis.

## Results

LPS-treated animals had endotoxin and AF monocyte chemoattractant protein (MCP)-1 concentrations that were significantly higher at 24 hours (immediately prior to rytvela administration) relative to values from Saline Group animals. Following rytvela administration, the average MCP-1 concentrations in the AF were significantly lower in the LPS + rytvela Group relative to in the LPS Group. In delivery samples, the expression of IL-1β in fetal skin was significantly lower in the LPS + rytvela Group compared to the LPS Group.

## Conclusion

A single dose of rytvela was associated with partial, modest inhibition in the expression of a panel of cytokines/chemokines in fetal tissues undergoing an active inflammatory response.

## Background

Preterm birth (PTB; delivery before 37 weeks of completed gestation) is a major cause of neonatal mortality and morbidity, with the greatest rates of death and significant disease seen in premature deliveries occurring prior to 32 weeks' gestation [1, 2]. PTB is a multifactorial syndrome [1, 2]. Many studies have investigated the role of intraamniotic inflammation in PTB; comparatively few studies have evaluated the efficacy of antenatal treatments for infection and inflammation-associated prematurity [1, 3, 4].

Interleukin (IL)-1 is a potent pro-inflammatory cytokine' [4, 5]. IL-1β concentration and bioactivity in amniotic fluid of women is related to preterm labor (PTL), infection [6], and PTB [7]. Intra-amniotic (IA) injection of 10 mg lipopolysaccharide (LPS) caused elevated concentrations of IL-1 in the amniotic fluid, chorioamnionitis, lung inflammation, and systemic inflammation in studies using preterm ovine model. Several studies have previously reported investigations into the efficacy of IL-1 targeting agents to prevent PTB and fetal injury [4, 5], using agents that competitively antagonize the IL-1 receptor and therefore likely block all downstream IL-1 signal transduction, including NF-κB activation [4]. This approach may risk interruption to important physiological processes such as cytoprotection and immune-surveillance [8].

Rytvela is a selective allosteric inhibitor of IL-1 receptor signalling [9, 10]. Allosteric antagonists interact with the receptor at a site remote from the orthosteric binding site, allowing the ligand to bind its receptor. They are biased signal inhibitors; i.e. they selectively inhibit some but not all intracellular signals of the receptor [11, 12]. A previous study showed rytvela selectively inhibited IL-1Racp downstream stress-associated protein kinase JNK, mitogen-activated protein kinases p38 and Rho/Rho GTPase/Rho-associated coiled-coil-protein kinase pathway, without affecting NF-κB activation [9]. Therefore, unlike other IL-1-targeting agents, rytvela can exert functional selectivity [11]. Accordingly, rytvela likely does not inhibit NF-κB [10] or suppress immune-surveillance. Due to its small size, rytvela is expected to have good

bioavailability [13]. Recent studies demonstrated that rytvela (1 mg/kg) reduced PTB and perinatal death, and reduced inflammation in the fetal brain, lung and colon of mice exposed to LPS [4, 9, 10]. No evidence of reproductive toxicity was detected and rytvela was shown to have a promising pharmacological profile [4, 5, 10].

It remains to be determined if rytvela allows for control of established intrauterine inflammation in extremely preterm fetal lambs–an established model for human fetal inflammation/infection. Making this determination is important because: i) patients in preterm labour are commonly diagnosed with intrauterine infection / inflammation after already presenting with symptoms; and ii) infection and inflammation are most commonly associated with early preterm delivery and fetal injury. In this experiment, rytvela was administered to extremely preterm ovine fetuses 24 hours after LPS exposure to simulate a potential clinical intervention scenario. While clinically rytvela would be given maternally, in this study direct fetal and intramniotic administration of rytvela was employed to ensure accurate intraamniotic dosing. We hypothesized that antenatal administration of rytvela treatment would reduce intra-amniotic and fetal inflammation.

## Methods

### Animals

Animal studies were approved by The University of Western Australia's Animal Ethics Committee (approval RA.3.100.1638). Ewes with a singleton fetus at 95 days of gestation (term is ~150 days) were fasted overnight before undergoing recovery surgery to place fetal jugular and intra-amniotic (IA) catheters, as previously reported [14]. After a 48-hour recovery, animals were randomized to one of the following groups (Fig 1): **i)** a single IA bolus of 2 ml saline 24 hours before both 2 ml IA and 2 ml fetal intravenous (IV) injection of saline (Saline Group, n = 7); **ii)** IA injection of 10 mg LPS from *Escherichia coli* (055:B5; Sigma Aldrich, St Louis, Missouri) in 2 ml saline 24 hours before 2 ml IA and 2 ml fetal IV injection of saline (LPS Group, n = 10); **3)** IA injection of LPS in 2 ml saline 24 hours before 0.3 mg/kg IA and 1 mg/kg fetal IV injection of rytvela, both in 2 ml saline vehicle (LPS + rytvela Group, n = 7). Rytvela was administered into the fetal circulation and amniotic cavity to target systemic and localized (amniotic fluid-exposed) inflammation, with dosing based on previous efficacious use in a

## Study design

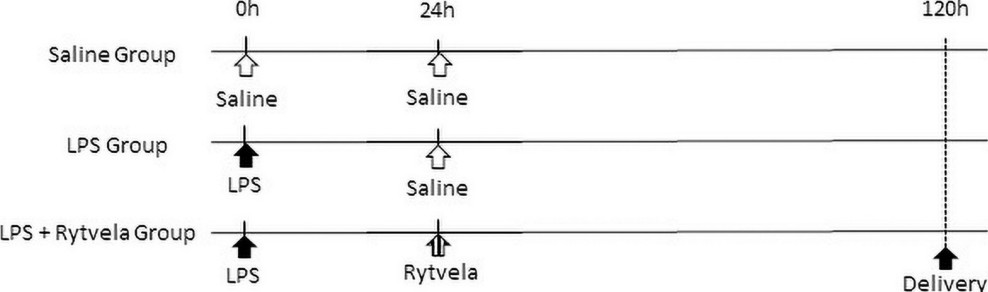

**Fig 1. Study design.** The LPS Group and the LPS + Rytvela Group animals received an IA injection of 10 mg LPS at 0 hour. The Saline Group animals received an equivalent volume of saline solution. The LPS + rytvela Group received both IA and fetal intravenous injection of rytvela at 24 hours after LPS exposure. The Saline Group and the LPS Group animals received an equivalent volume of saline solution. IA, intra-amniotic; LPS, lipopolysaccharide; MCP, monocyte chemoattractant protein.

small animal model [5, 9, 10]. Animal wellbeing was monitored daily, with free access to food and water.

Amniotic fluid (AF) was serially sampled from amniotic catheters at 0, 12, 24, 48, 72, and 120 hours after the first administration of either saline or LPS. Animals were euthanized with an intravenous bolus (160 mg/kg) of pentobarbital at 120 hours. Fetal cord blood was collected at delivery for blood chemistry and ELISA analysis. Fetal tissues were snap frozen in liquid nitrogen for protein or messenger ribonuclease acid (mRNA) expression analyses. For histological analyses, fetal tissue was placed in cassettes and fixed in 10% neutral buffered formalin before being processed by paraffin embedding.

## Endotoxin quantification

Endotoxin concentrations in amniotic fluid were measured by using a Pierce™ Chromogenic Endotoxin Quant Kit (Thermo Fisher, A39552), according to manufacturer's instructions. Each plate was read at 405 nm on a Spectramax plate reader (Molecular Devices LLC, Sunnyvale, CA).

## Enzyme-linked inmmunosorbent assay

Quantification of the cytokine IL-1 in amniotic fluid, cord blood at delivery, and fetal lung and IL-8 in amniotic fluid and cord plasma were measured by using a Sheep ELISA Kits (CUSA-BIO, CSB-E10115Sh), according to manufacturer's protocols. Each plate was read at 450 nm on a Spectramax plate reader (Molecular Devices LLC, Sunnyvale, CA). The software package Curve Expert, version 1.4 (CUSABIO), was used to make a standard curve. Similarly, quantification of the chemokine monocyte chemoattractant protein (MCP)-1 and cytokine tumor necrosis factor (TNF)-α in amniotic fluid and cord plasma were measured by using a Kingfisher Biotech (Saint Paul, MN) ELISA Development Kits, according to the manufacture's protocol. Each plate was read at 450 nm on a Spectramax plate reader (Molecular Devices LLC, Sunnyvale, CA).

## Tissue RNA isolation

DNA-free total RNA was isolated from frozen fetal tissues (lung right lower lobe, internal groin skin, colon, chorioamnion and brain cortex) using RNeasy® Plus Mini Kit (QIAGEN, Hilden, Germany), according to the manufacture's protocol. The concentration and quality of extracted RNA was quantified with a Qubit 2.0 fluorometer (Life Technologies, Carlsbad, CA).

## Quantitative polymerase chain reaction

Ovine-specific PCR primers and hydrolysis probes for IL-1β, IL-6, TNF-α and MCP-1 (Thermofisher) were used to perform quantitative PCR reactions on RNA from fetal lung, skin, colon, chorioamnion and brain cortex tissue. Fetal lung, skin, chorioamnion and colon were selected for analysis as they are exposed to the amniotic fluid directly or via swallowing and have been shown to initiative a pro-inflammatory response following IA LPS administration. Inflammation of the fetal brain in the setting of preterm birth is of significant interest given its association with neuro-developmental damage.

Reactions were conducted using a Step One Real-Time PCR system and probe/primer sets from Life Technologies with RNA normalised to 25 ng/μl. Reaction cycling conditions were as follows: $1 \times 15$ min reverse transcription [50˚C], $1 \times 20$ seconds initial denaturation [95˚C], followed by 40 cycles of 3 seconds denaturation [95˚C] and 30 seconds annealing [60˚C]. All reactions were performed in 96 well fast plates on a ViiA7 real-time PCR themocycler (Life

Technologies) and in triplicate. Averaged quantitation cycle (Cq) values were normalized against averaged Cq values of ribosomal protein 18s. Data were processed to generate fold changes using a 2-$^{ddCq}$ method and were tested for significance.

## Histology of fetal lung and chorion-amnion

The right upper lobe of each lung was inflation fixed with 4% paraformaldehyde at 30 cmH$_2$O pressure. The inflammation of both fetal lung and chorion-amnion was evaluated using hematoxylin and eosin (H&E)-stained sections and was graded in a blinded fashion by a scoring methods follows: Fetal lung was assessed using three 5-μm sections from each animal as 0 (no inflammatory cells), 1 (a few inflammatory cells), 2 (moderate influx of cells) or 3 (extensive influx of inflammatory cells), as previously published [15, 16].

Chorioamnion rolls were fixed in 4% paraformaldehyde. Chorioamnionitis was classified using a modified grading system based on that reported by Redline et. al. [17]. Histological staining was used to determine the localization of inflammatory cells and (where present) tissue damage. The grade evaluation also identifies the intensity of the inflammatory cell infiltration [17, 18]. The chorioamnion of each animal was graded based on this classification as previously reported [19]: 0 (no chorioamnionitis), 1 (localization: within choriodecidua; intensity: individual or small clusters of inflammation cell), 2 (localization: within choriodecidua; intensity: the three or more chorionic microabscesses which are defined as inflammatory cell $\geq 10 \times 20$ cells), 3 (localization: choriodecidua and/or amnion, intensity: same as score = 2), 4 (localization: necrosis of the amniotic epithelium; intensity: same as score = 2). Six 5-μm sections from each animal were evaluated at random and an average score was calculated for each animal.

## Fluorescent immunohistochemical analysis of fetal brain

The left hemisphere was fixed in 4% paraformaldehyde and sectioned in 4-mm serial section (5 serial sections in total) along the coronal plane from the anterior to posterior sections according to the method of Banker and Larroche [20]. The second serial section was selected to be embedded in paraffin and stained using following primary antibodies: glial fibrillary acidic protein (GFAP) (1:1000, ab7260, Abcam), ionized calcium-binding adapter molecule 1 (Iba-1) (1:500, ab178846, Abcam), oligodendrocyte transcription factor 2 (Olig-2) (1:2000, NBP1-28667, Novusbio) and neuronal nuclei (NeuN) (1:750, ab128886, Abcam). Alexa Fluor® 488 (1:10000, ab150077, Abcam) was used as secondary antibody and nuclei were stained with DAPI (F6057, Sigma-Aldrich). Microphotographs were captured using a confocal microscope (Nikon A1Si). Five random, non-overlapping fields per section were assessed for each antigen target. Numbers of GFAP-, Iba-1-, Olig-2- and NeuN-positive / DAPI-positive cells in subcortical white matter were counted manually by single investigator blinded to treatment.

## Western blot analysis of fetal brain tissue

Proteins from the right hemisphere cortex were extracted using RIPA lysis buffer (Thermo Fisher, 89901) and a cocktail of protease inhibitors (Merck, 04693116001) following cold maceration on a Precellys 24 tissue homogenizer. Protein concentration was quantified with a Pierce™ Rapid Gold BCA Protein Assay Kit (Thermo Fisher, A53226). 15 μg of total protein per well was resolved using a NuPAGE™ 10% Bris-Tris Gel and electrotransferred to PVDF membrane according to manufacturer's instructions. Membranes were blocked with blocking buffer (Thermo Fisher, 37565) for 30 minutes at room temperature and incubated with No-Stain™ Protein labeling Reagent (A44717, Thermo Fisher) according to the manufacture's

instructions. Membranes were then incubated overnight at 4˚C on a rocker platform with following primary antibodies: GFAP (1:5000, ab7260, Abcam), Iba-1 (1:1000, ab178846, Abcam), Olig-2 (1:3000, NBP1-28667, Novusbio) and NeuN (1:1000, ab128886, Abcam). Western blot enhancer (SuperSignal™ Western Blot Enhancer, 46640, Thermo Fisher) was used for Olig-2 according to the manufacture's protocol. Membranes were washed for 30 minutes in PBS-Tween before being incubated with Alexa Fluor® 488 (1:10000, ab150077, Abcam) for 1 hour at room temperature. Membranes was analyzed with iBright™ FL1000 imaging system (Thermo Fisher). Each band volume was normalized against the total protein amounts and each membrane was compensated by the total protein amounts of quality control animal.

## Hematology

Complete blood counts (CBC) were performed by VetPath Laboratory Services using an automated Coulter counter customized for sheep (Ascot, Perth, Western Australia).

## Statistical analysis

Data were analysed for distribution, variance and skew. All values are expressed as either mean ± one standard deviation or median ± IQR. Analyses were performed using IBM SPSS Statistics for Windows, version 20.0 (IBM Corp. Armonk, NY.). Paired comparisons were performed with $t$-tests or Mann-Whitney tests. The comparison of more than two groups was tested by one-way ANOVA analysis of variance with multiple post-hoc comparisons used by Tukey's t or Kruskal-Wallis post-hoc analyses. Significance was accepted at $p < 0.05$.

# Results

There were no significant differences in birthweight, cord blood pH, $pCO_2$, or lactate at delivery between the three treatment groups (Table 1). No increase of fetal edema or ascites as a result of lipopolysaccharide exposure or rytvela administration was detected. Variability in the group sizes used to analyse time-course samples obtained by catheter sampling reflect changes in fetal positioning which impact catheter patency.

## Endotoxin quantification

Endotoxin concentrations from the LPS-treated Groups (the LPS Group and the LPS + rytvela Group) at 10 hours after LPS administration in AF were significantly increased compared to AF samples from the Saline Group. (Saline Group vs. LPS-treated Groups p<0.000) (Fig 2A).

## ELISA

**Pre-intervention analysis.** We evaluated the effect of IA administration of LPS or saline from 0 to 24 hours (before randomization to treatment by either saline or rytvela). No difference was detected in AF concentrations in IL-1β, IL-8, MCP-1 or TNF-α at 0 hours between the Saline-treated and LPS-treated animals. AF MCP-1 concentrations in the LPS-treated

**Table 1. Delivery data.**

| Group | Sex (M/F) | Delivery Wt. (kg) | Arterial fetal arterial cord blood | | |
| | | | pH | pCO2 (mmHg) | Lactate (mmol/L) |
|---|---|---|---|---|---|
| Saline Group (n = 7) | 3/4 | 1.1 ± 0.1 | 7.1 ± 0.1 | 78.0 ± 8.5 | 4.4 ± 0.9 |
| LPS Group (n = 10) | 6/4 | 1.2 ± 0.2 | 7.1 ± 0.1 | 75.8 ± 9.1 | 4.1 ± 1.3 |
| LPS + Rytvela Group (n = 7) | 7/0 | 1.2 ± 0.1 | 7.1 ± 0.1 | 81.3 ± 13.9 | 4.4 ± 1.1 |

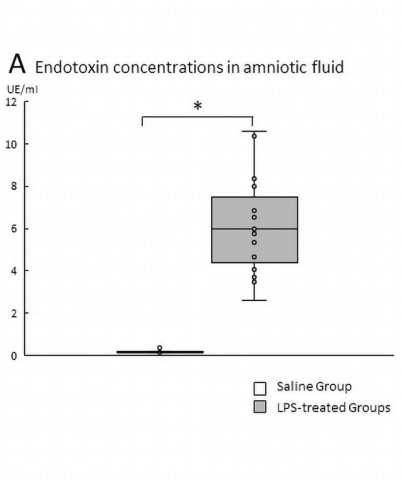

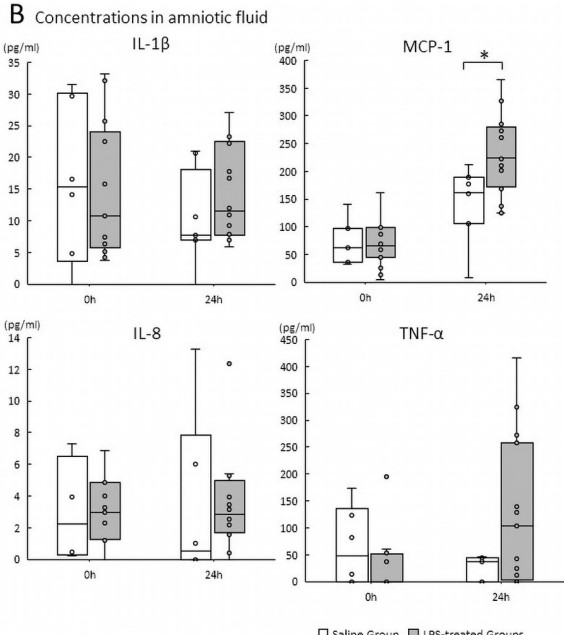

**Fig 2.** A. Endotoxin concentrations in amniotic fluid. Plots show median, 25th, and 75th percentile ranges of AF endotoxin concentrations at 24 hours post-administration as boxes and 5th to 95th percentile as error bars. *p < .05 vs. the Saline Group. n = 6–10 animals/group. AF, amniotic fluid; LPS, lipopolysaccharide. B. Cytokines/chemokine concentrations in amniotic fluid from 0 to 24 hours after either saline or LPS. A, amniotic fluid IL-1β, IL-8, MCP-1, and TNF-α concentrations. *p < .05 between the Saline Group and the LPS-treated Groups (LPS Group and LPS + rytvela Group). Plots show median, 25th, and 75th percentile ranges as boxes and 5th to 95th percentile as error bars. n = 4–8 animals/group. AF, amniotic fluid; LPS, lipopolysaccharide; ELISA, Enzyme-linked inmmunosorbent assay.

animals increased significantly at 24 hours compared with Saline-treated animals (p = 0.016) (Fig 2B). No significant differences were detected in IL-1β, IL-8 and TNF-α concentrations in AF between Saline-treated animals and LPS-treated animals at 24 hours.

**Post-intervention analysis.** We then compared the average concentration of cytokines/chemokines at time points between the LPS Group and the LPS + rytvela Group (Fig 3). The average MCP-1 concentration in AF from the LPS + rytvela Group decreased significantly compared to the LPS Group (p = 0.002) (Fig 3B). There were no significant differences in IL-1β, IL-8 and TNF-α AF concentrations (Fig 3A, 3C and 3D).

**At delivery.** There was a significant difference in IL-1β protein concentration in fetal lung tissue at delivery between the Saline Group and the LPS Group (p = 0.023). No difference was detected between the Saline Group and the LPS + rytvela Group (p = 0.129) (Fig 4).

## qPCR

Results are shown in Fig 5. Statistically significant increases were detected in fetal lung IL-1β mRNA in both the LPS and LPS + rytvela Groups relative to the Saline Group animals (Saline Group vs. LPS Group p = 0.043; Saline Group vs. LPS + rytvela Group p = 0.02). TNF-α mRNA in the fetal lung was significantly increased in the LPS + rytvela Group animals compared to the Saline Group animals (Saline Group vs. LPS + rytvela Group p = 0.029) (Fig 6A). IL-1β mRNA in the fetal skin increased significantly in the LPS Group in comparison to the Saline Group (Saline Group vs. LPS Group p = 0.004). Moreover, IL-1β mRNA concentration in fetal skin was significantly lower in the LPS + rytvela Group compared to the LPS Group (LPS Group vs. LPS + Rytvela Group p = 0.002) (Fig 6B).

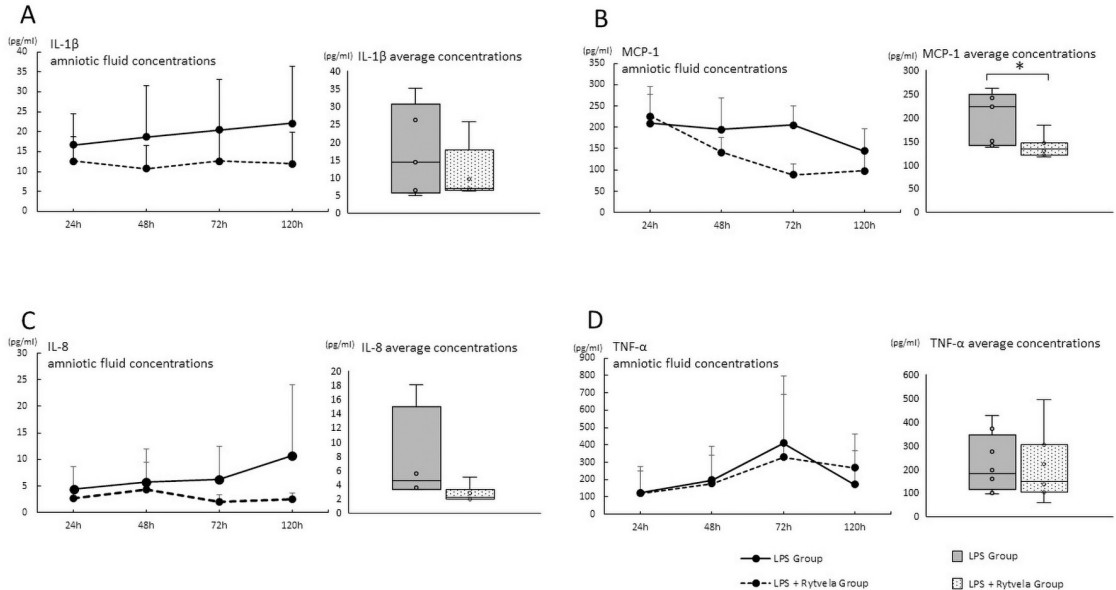

**Fig 3. Cytokines/chemokine concentrations in amniotic fluid after the administration of either saline or rytvela).** Cytokines/chemokine concentration in AF at each time point plotted as mean ± SD. Each group's average concentration is shown in bar charts. **A**, IL-1β; **B**, MCP-1; **C**, IL-8; **D**, TNF-α. The average concentration was compared between the LPS Group and the LPS + rytvela Group. *p < .05 between the LPS Group and the LPS + rytvela Group. n = 3–7 animals/group. AF, amniotic fluid; SD, standard deviation; IL, Interleukin; MCP, monocyte chemoattractant protein; TNF, tumor necrosis factor.

The LPS Group had a significant increase in fetal skin TNF-α mRNA concentration compared with the Saline Group animals (Saline Group vs. LPS Group p = 0.016) (Fig 6B). Similar to fetal lung and skin, there were significant increases in IL-1β mRNA in the colon of LPS Group animals relative to the Saline Group animals (Saline Group vs. LPS Group p = 0.045) (Fig 6C). No differences were identified in TNF-α mRNA concentration in colon tissue between the three groups.

There were no significant differences in fetal chorioamnion in IL-1β mRNA concentration among the three groups; however, significant increases were detected in TNF-α mRNA concentration in the chorion-amnion of both the LPS Group and the LPS + rytvela Group animals relative to the Saline Group animals (Saline Group vs. LPS Group p = 0.01; Saline Group vs. LPS + Rytvela Group p = 0.03) (Fig 6D). There were no differences in IL-1β, TNF-α, IL-6 or MCP-1 mRNA detected in fetal brain (cortex), among the three groups (Fig 6E).

## Histology and fluorescent immunohistochemistry

Inflammation in the fetal lung and chorioamnion was evaluated by blind scoring of inflammatory cells by a single investigator. Inflammatory cell infiltration was detected in fetal lung in both the LPS Group and the LPS + rytvela Group; in contrast, few or no inflammatory cells were detected in the Saline Group animals (Fig 5B to 5D for representative images). The inflammation score of the LPS Group was significantly higher than that of the Saline Group, while there was no significant difference between the Saline Group and the LPS + rytvela Group animals (Saline Group vs. LPS Group p = 0.005; Saline Group vs. LPS + rytvela Group p = 0.142) (Fig 5A).

Similarly, inflammatory cells were detected in the chorion-amnion of both the LPS Group and the LPS + rytvela Group (Fig 7B to 7D for representative images). The inflammation score was significantly increased in the LPS Group relative to the Saline Group (Saline Group vs.

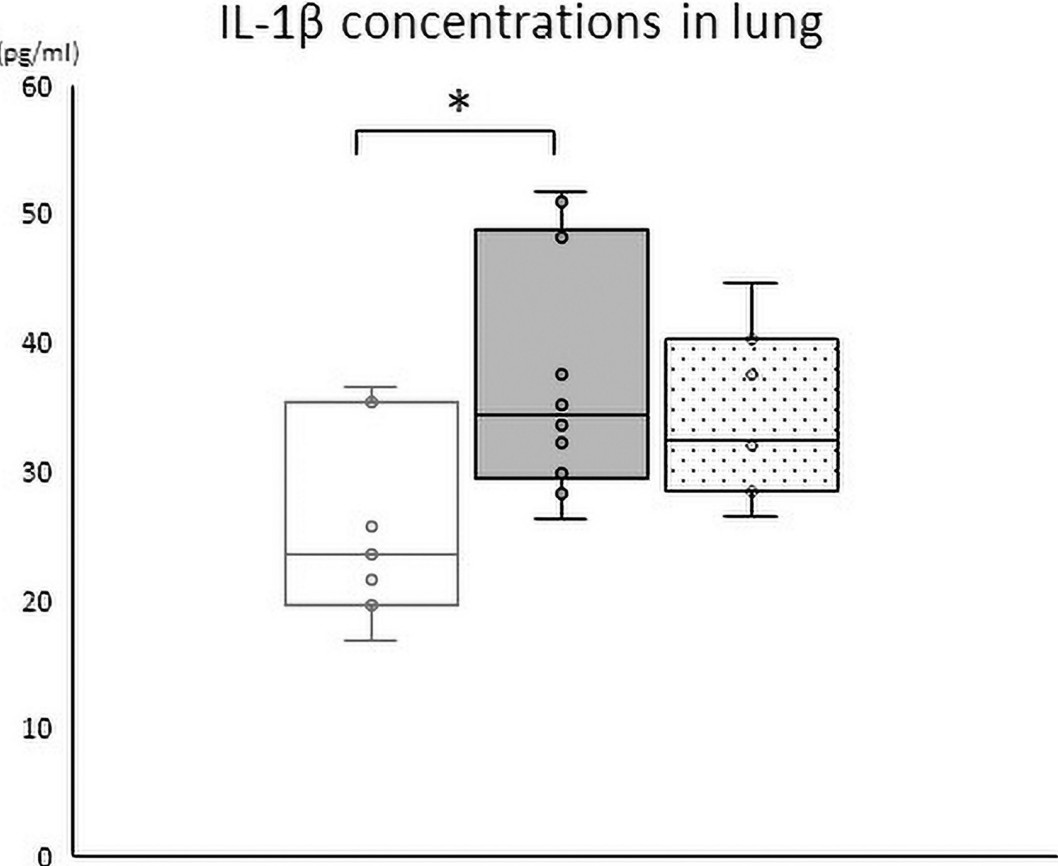

**Fig 4. IL-1β protein concentrations in fetal lung at delivery.** *p < .05 vs. the Saline Group. Plots show median, 25th, and 75th percentile ranges as boxes and 5th to 95th percentile as error bars. n = 7–10 animals/group IL, Interleukin.

LPS Group p = 0.006). The inflammation score in the LPS + rytvela Group was comparable to that of the Saline Group (Saline Group vs. LPS + Rytvela Group p = 0.23) (Fig 7A).

Fluorescent immunohistochemistry and western blotting was performed to assess fetal brain injury. Previous studies showed the reaction of fetal brain exposed to LPS are activated microglia and astrocyte, maturation arrest or death of oligodendrocyte, and axonal loss or death of neurons [21–23]. Therefore, as for analysis of the cortex, inflammation/injury was assessed by counting Iba-1-, GFAP-, Olig-2- and NeuN-positive / DAPI positive cells in sub-cortical white matter. There were no significant differences in the numbers of Iba-1-, GFAP-, Olig-2-, and NeuN-positive / DAPI positive cells in subcortical white matter among the three groups. (Fig 8A).

## Fluorescent western blotting

Results are shown in Fig 8B. Iba-1, GFAP, Olig-2 and NeuN specific bands (green bands) were observed and each band volume was normalized to total protein amounts (blue bands). There

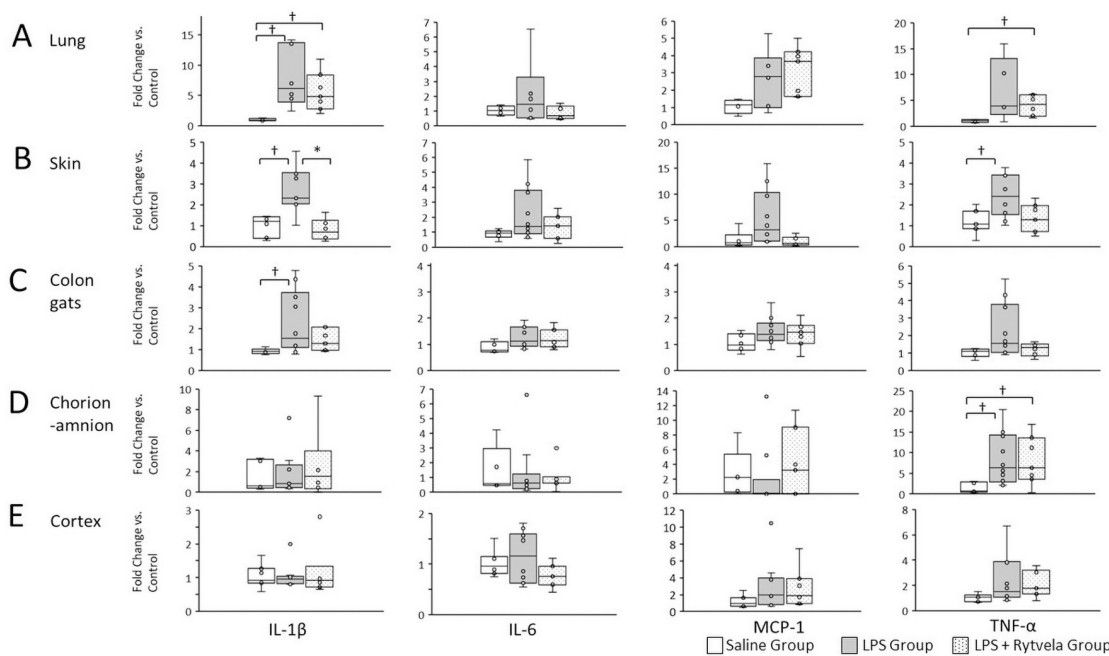

**Fig 5. Histological assessment of fetal lung. A**, Plots show median, 25th, and 75th percentile ranges as boxes and 5th to 95th percentile as error bars. LPS Group increased inflammation score in fetal lung tissue relative to Saline Group. *p < .05 between the Saline Group and the LPS Group. Representative sections of tissue from **B**, Saline Group (score = 0); **C**, LPS Group (score = 2); **D**, LPS + rytvela Group (score = 1). Inflammatory cells are indicated by *arrows*. *Scale bar*: 100 μm. n = 7–10 animals/group. LPS, lipopolysaccharide.

was no significant differences of the band volume of Iba-1, GFAP, Olig-2 and NeuN in fetal cortex (Fig 8B) between groups.

## Haematology

There was a significant increase in the total white blood cell count at delivery in both the LPS Group and the LPS + rytvela Group fetuses relative to the Saline Group animals (Saline Group vs. LPS Group p = 0.036; Saline Group vs. LPS + rytvela Group p = 0.029) (Fig 9A). Similarly, the ratio of fetal neutrophils to total white blood cells at delivery was significantly increased in both the LPS Group and the LPS + rytvela Group animals compared with the Saline Group animals (Saline Group vs. LPS Group p = 0.001; Saline Group vs. LPS + Rytvela Group p = 0.001). The ratio of lymphocytes to total white blood cells was significantly decreased in both the LPS Group and the LPS + rytvela Group animals when compared with the Saline Group animals (Saline Group vs. LPS Group p = 0.003; Saline Group vs. LPS + rytvela Group p = 0.001) (Fig 9B and 9C).

## Discussion

### Primary findings

These data show that, in extremely preterm sheep fetuses, the introduction of LPS to the amniotic cavity induced chorioamnionitis and increased tissue inflammation. The combined intraamniotic and fetal intravenous administration of rytvela was associated with a modest partial reduction in fetal and intraamniotic inflammation even 24 hours following LPS exposure, relative to treatment with saline only. As such, with further optimization (dose,

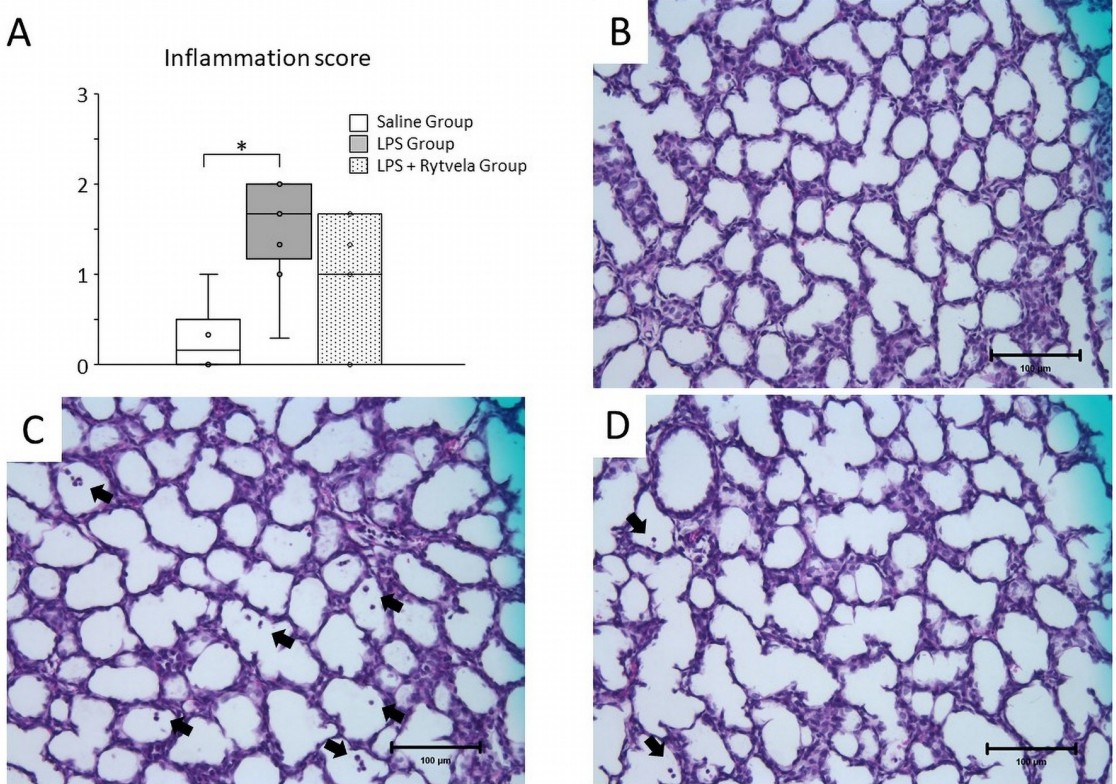

**Fig 6. Messenger RNA expression in at delivery.** Relative expression of mRNA transcripts for IL-1β, IL-6, MCP-1, and TNF-α in **A**, fetal lung; **B**, skin; **C**, colon; **D**, chorioamnion; **E**, cortex. †p < .05 vs the Saline Group. *p < .05 between the LPS Group and the LPS + rytvela Group. Plots show median, 25th, and 75th percentile ranges as boxes and 5th to 95th percentile as error bars. n = 7–10 animals/group. IL, Interleukin; MCP, monocyte chemoattractant protein; TNF, tumor necrosis factor.

timing, route of administration) of rytvela may show promise as treatment for intra-amniotic inflammation.

## Results

**Amniotic fluid inflammation.** LPS-associated increases in TNF-α expression were noted in fetal tissues, although not in the AF as previously reported [24]. Similarly, differences in IL-1β expression were only detected in fetal tissue. Previous studies found that IL-1β induced MCP-1 expression in human cells via both NF-κB-mediated and alternative signaling pathways [25], while significantly increased MCP-1 concentration in the AF of women were associated with evidence of intra-amniotic infection [26]. Moreover, studies have also shown that the concentration of MCP-1 in AF from women who delivered preterm without apparent intra-amniotic infection was significantly higher than those who delivered at term delivery [26, 27]. Therefore, MCP-1 is a useful biomarker to the existence of intraamniotic infection regardless of whether the antigen is identified or not. The amount of MCP-1 protein detected was significantly elevated in the LPS-treated animals compared with the Saline Group. Moreover, in the LPS + rytvela Group, MCP-1 average AF concentrations were significantly decreased, relative to values in the LPS Group, suggesting a partial inhibition of inflammation in response to rytvela treatment. We and others have previously demonstrated that extended exposure to intrauterine inflammation negatively affects fetal outcome [15, 21, 28]. As such, reduction of intraamniotic inflammation may reduce the risk and/or severity of fetal injury.

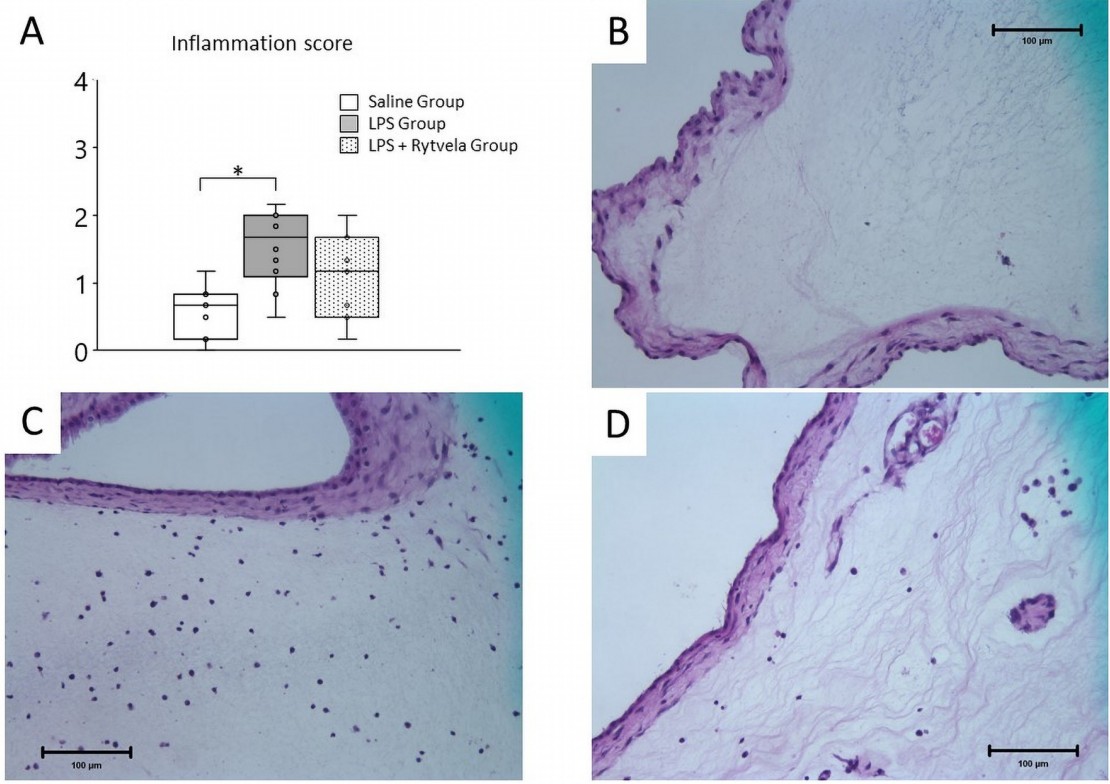

**Fig 7. Histological assessment of chorioamnion. A**, Plots show median, 25th, and 75th percentile ranges as boxes and 5th to 95th percentiles as error bars. The LPS Group increased inflammation score in chorion-amnion tissue relative to the Saline Group. *p < .05 between the Saline Group and the LPS Group. Representative sections of tissue from **B**, Saline Group (score = 0); **C**, LPS Group (score = 2); **D**, LPS + rytvela Group (score = 1). Scale bar: 100 μm. n = 7–10 animals/group. LPS, lipopolysaccharide.

No statistically significant increase in either IL-1β or IL-8 was detected in the AF between LPS and non-LPS groups prior to treatment. Although this reason for this observation is not known exactly, it is possible that the timing of sampling was too early or too late to optimally capture increase expression. It is also possible that an increase in baseline inflammation deriving from fetal and maternal surgery increased the noise in the analysis, partly masking a difference in AF inflammation between groups.

**Fetal tissue inflammation.** Animals in the LPS Group had significant increases in IL-1β mRNA expression in all three organs assessed (skin, lung and colon) that were exposed to the AF (and thus directly to LPS), compared to the Saline Group, whereas there were no differences detected between the LPS + rytvela Group and the Saline Group animals, suggesting some degree of resolution of inflammation signaling by rytvela. Similarly, ELISA results showed that IL-1β concentrations in fetal lung from the LPS Group were significantly increased compared to the Saline Group, while no difference was found between the LPS + rytvela Group and the Saline Group.

The LPS Group and the LPS + rytvela Group animals both had significantly increased WBC counts and altered neutrophil and lymphocyte ratios. Histopathological evaluation of the fetal lung and the chorioamnion also showed the inflammation score increased significantly in the LPS Group compared to the Saline Group, while there was no significant differences between the LPS + rytvela Group and the Saline Group–although the range of scores therein was increased. Moreover, IL-1β mRNA expression in the skin of fetuses in the LPS + rytvela Group

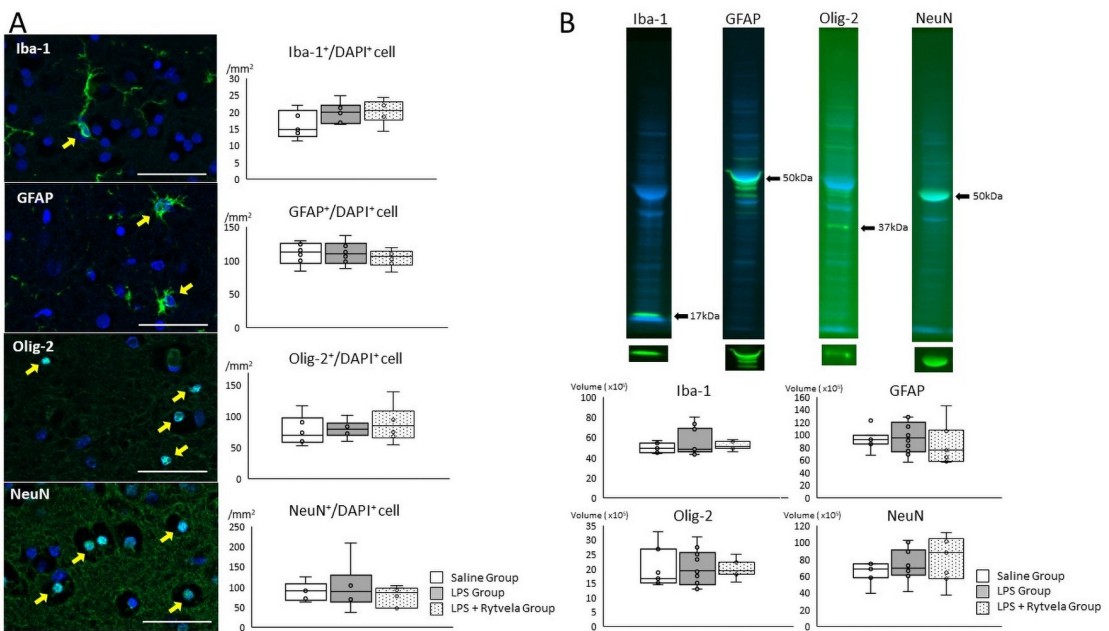

**Fig 8. Fluorescent immunohistochemistry and western blotting assessment of fetal brain. A**, Representative photomicrographs show Iba-1-, GFAP-, Olig-2- and NeuN- positive (green) / DAPI positive (blue) cells in subcortical white matter of the Saline Group animals. Positive cells are indicated by *arrows*. *Scale bar*: 50 μm. Box plots show the numbers of counting Iba-1-, GFAP-, Olig-2- and NeuN- positive / DAPI positive cells of three groups. **B**, Representative images show Iba-1, GFAP, Olig-2 and NeuN specific band (green) and total protein stained band (blue) of the quality control animal (Saline Group animal). Box plot show the band volume of the each antibody against the total protein amounts. Plots show median, 25th, and 75th percentile ranges as boxes and 5th to 95th percentile as error bars. n = 6–8 animals/group. Iba-1, ionized calcium-binding adapter molecule 1; GFAP, glial fibrillary acidic protein; Olig-2, oligodendrocyte transcription factor 2; NeuN, neuronal nuclei.

was significantly lower than that of animals from the LPS Group. Previous studies have identified the fetal skin is as a pro-inflammatory organ following intrauterine endotoxin exposure in the preterm sheep [29, 30]. We previously demonstrated that selective exposure of fetal skin + amnion to LPS induce acute systemic inflammation [31] and significant elevation of mRNA expression for IL-1β in fetal skin persisted till 15 days after LPS exposure [32], while that in

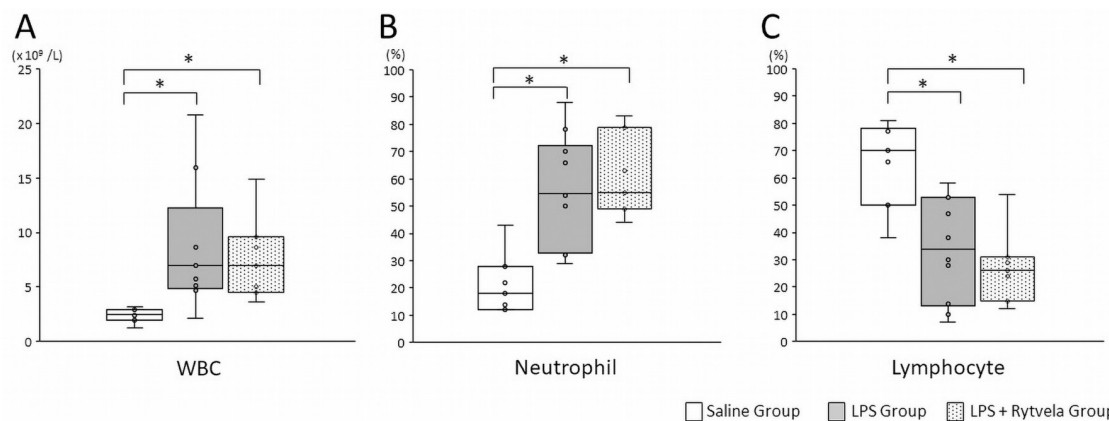

**Fig 9. Haematological assessments.** Plots show median, 25th, and 75th percentile ranges as boxes and 5th to 95th percentile as error bars. Intra-amniotic injection of LPS affected **A**, total white blood cell (WBC) counts; **B**, ratio of neutrophil; **C**, ratio of lymphocyte from cord blood. LPS, lipopolysaccharide. n = 7–9 animals/group.

fetal lung persisted till only 2days after LPS exposure [15]. As such, significant decrease of IL-1β mRNA expression in rytvela-treated fetal skin can be taken as suggestive of a beneficial effect of rytvela.

Taken together, these data suggest a modest reduction in fetal inflammation associated with the administration of rytvela. Given the exploratory nature of the dosing employed in this study and the persistent half-life of endotoxin in the intraamniotic environment, more frequent administration of rytvela may offer improved dampening of inflammatory responses.

## Clinical implications

It is widely accepted that intra-amniotic infection can cause acute neonatal mortality as well as long-term adverse outcomes. Currently, if the intra-amniotic infection and inflammation are suspected on the basis of a patient presenting with maternal fever, vaginal discharge, fetal distress or PTL, responses are broadly limited to the use of antibiotics and/or the management of delivery. A means of rapidly resolving intrauterine inflammation would be of potential use and likely to improve neonatal outcomes.

There have been several studies investigating the efficacy of IL-1 targeting agents to prevent PTB and fetal injury [4, 5, 9, 10]. The agents used in these previous studies all antagonized IL-1 actions competitively and blocked all downstream signal transduction [4], including that of NF-κB; this is thought to affect immune-surveillance and may result in an increased risk of fetal and maternal infection [8, 33]. Therefore, the functional selectivity of rytvela may allow for a reduction in the risk of adverse effects.

Agents such as rytvela regulate inflammatory responses but do not resolve the underlying infection or sequester pro-inflammatory antigen (in contrast to antibiotics such as polymyxin-B). These agents also have limited half-lives *in vivo*. Thus, any efficacious treatment regimen will likely be multi-faceted to allow for resolution of infection, inflammation signaling, and inhibition of further inflammatory stimulation. Additionally, in clinical practice there is a lack of effective and established diagnostic tests for women who are at risk of intra-amniotic infection leading to PTB, especially before 32 weeks gestation. As such, the successful application of agents such as rytvela (in combination with antimicrobial agents) will likely require a tandem improvement in the reliability of diagnostic tests for women at risk of intrauterine infection and PTL.

## Research implications

This study suggests potential efficacy of rytvela for treatment of intraamniotic infection during pregnancy. It is unclear whether IA or IV administration alone was sufficient for transient resolution of inflammation, or whether treatment of both the fetus and the amniotic fluid is necessary. As noted above, a significant amount of work is still required to optimize dose, timing, and duration of rytvela treatment. Ongoing development of this approach should focus on enhancing efficacy without additional adverse side-effects, in both sterile inflammation (i.e. LPS) and in model systems wherein an active infection is established.

## Limitations

Data presented herein should be interpreted in light of several considerations. Firstly, as a surgical model, the impact of LPS and peptide exposures may be somewhat masked by background inflammation–with the surgery likely causing a higher background level of inflammation. Secondly, the limited number of animals used in this study (by virtue of the intensive experimental design) combined with the wide group standard deviations of the ELISA results mean that these data should be interpreted with appropriate caution. Thirdly, as

the dosing regimen adopted was exploratory and designed to inform subsequent experiments it should not be considered optimal. Lastly, a lack of brain inflammation in the present study restricts our ability to assess the potential neuro-protective benefits of rytvela. The reasons for this are unclear, but may relate to the early gestational age of the fetuses used in this study, the comparatively short period between LPS exposure and sample collection, and the region of brain assessed.

A number of factors including the inability of LPS to cross cell-cell barriers, a lack of multi-antigen stimulation, and an absence of viable infection mean that host responses to LPS likely differ from viable microorganisms. The half-life of LPS in the amniotic fluid is 1.7 days [34] and inflammation is gradually reduced without any intervention, which may not reflect clinical intra-amniotic infection. Despite this limitation, results showing the rytvela administration even 24 h following LPS exposure was associated with decreased inflammation in the AF and fetal tissues is still significant.

Given the intensive nature of studies employing the chronically catheterized sheep model, we were able to select only one dose of rytvela, based on the mouse literature, and were restricted in terms of our group sizes. The timing of rytvela administration (24h after LPS) was designed to allow capture a broad cross-section of protein, molecular and histological data from multiple tissues over a 5-day time course. Given the ontology of intrauterine inflammation in the sheep model, studies targeting individual tissues (especially the brain) and signals at specified time points will be required for more granular assessments. Irrespective, the data presented herein will inform future studies testing other rytvela dosing levels and timing relative to LPS administration to optimize both. These will also compare the efficacy of fetal IV-only vs. amniotic fluid-only administration of rytvela and need to assess the adverse effect using rytvela-only model.

## Conclusions

We report that administration of the peptide rytvela partially reduced fetal and amniotic fluid inflammation following IA LPS challenge in an ovine model of chorioamnionitis. Given that dosing was exploratory and inflammation already established, these findings may be cautiously viewed as promising. Additional work focusing on optimizing dosing regimen may allow for a more comprehensive resolution of inflammation associated with preterm birth and fetal injury.

## Supporting information

**S1 Raw images.**
(PDF)

## Acknowledgments

We would like to thank Sara and Andrew Ritchie (Icon Agriculture, Darkan, Western Australia) for their expertise and providing date-mated sheep for this study, and Dr Christiane Quiniou for providing and validating the rytvela used in this study and for valuable advice on its storage and handling.

## Author Contributions

**Conceptualization:** Yuki Takahashi, Masatoshi Saito, Haruo Usuda, Tsukasa Takahashi, Erin L. Fee, Sylvain Chemtob, Jeffrey Keelan, David Olson, John P. Newnham, Matthew W. Kemp.

**Data curation:** Yuki Takahashi.

**Formal analysis:** Yuki Takahashi, Masatoshi Saito, Haruo Usuda, Tsukasa Takahashi.

**Funding acquisition:** Sylvain Chemtob, Jeffrey Keelan, David Olson, Matthew W. Kemp.

**Investigation:** Yuki Takahashi, Masatoshi Saito, Haruo Usuda, Tsukasa Takahashi, Shimpei Watanabe, Takushi Hanita, Shinichi Sato, Yusaku Kumagai, Shota Koshinami, Hideyuki Ikeda, Sean Carter, Erin L. Fee, Lucy Furfaro, Sylvain Chemtob, Alan H. Jobe, Matthew W. Kemp.

**Methodology:** Yuki Takahashi, Masatoshi Saito, Haruo Usuda, Tsukasa Takahashi, Shimpei Watanabe, Takushi Hanita, Shinichi Sato, Yusaku Kumagai, Shota Koshinami, Hideyuki Ikeda, Sean Carter, Erin L. Fee, Lucy Furfaro, Sylvain Chemtob, John P. Newnham, Matthew W. Kemp.

**Project administration:** Yuki Takahashi, Haruo Usuda, Erin L. Fee, Jeffrey Keelan, David Olson, Alan H. Jobe, Matthew W. Kemp.

**Resources:** Matthew W. Kemp.

**Supervision:** Masatoshi Saito, Haruo Usuda, Tsukasa Takahashi, Shimpei Watanabe, Takushi Hanita, Shinichi Sato, Yusaku Kumagai, Nobuo Yaegashi, John P. Newnham, Alan H. Jobe, Matthew W. Kemp.

**Writing – original draft:** Yuki Takahashi, Masatoshi Saito, Tsukasa Takahashi, Matthew W. Kemp.

**Writing – review & editing:** Yuki Takahashi, Masatoshi Saito, Haruo Usuda, Tsukasa Takahashi, Shimpei Watanabe, Takushi Hanita, Shinichi Sato, Yusaku Kumagai, Shota Koshinami, Hideyuki Ikeda, Sean Carter, Erin L. Fee, Lucy Furfaro, Sylvain Chemtob, Jeffrey Keelan, David Olson, Nobuo Yaegashi, John P. Newnham, Alan H. Jobe, Matthew W. Kemp.

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
