## [Decision Letter · Decision Letter 0]

3 Jun 2021

PONE-D-21-12998

Direct administration of the non-competitive interleukin-1 receptor antagonist rytvela transiently reduced intrauterine inflammation in an extremely preterm sheep model of chorioamnionitis

PLOS ONE

Dear Dr. Kemp,

Thank you very much for submitting your manuscript PONE-D-21-12998 "Direct administration of the non-competitive interleukin-1 receptor antagonist rytvela transiently reduced intrauterine inflammation in an extremely preterm sheep model of chorioamnionitis" to PLOS ONE. Your manuscript has been assessed by two expert reviewers, and the full text of their reviewer reports is appended below, for your information. The reviewers found merit in your study, and have recommended the further processing of your report at PLOS ONE subject to satisfactory responses to a number of minor concerns that they have outlined in their reviewer reports. Thus, I am returning your manuscript to you for MINOR REVISION.

In the preparation of your revised manuscript, please consider and respond to all concerns raised by the two expert reviewers in their reviewer reports. These concerns include a request for further information about (i) the rationalization of the dose, (ii) LPS responsiveness of selected genes, (iii) n number and biological replicates, and their declaration in the manuscript, (iv) the translational relevance of the skin findings, (v) that rational for the examination of neuronal markers; (vi) the relationship between IL-1beta and MCP-1; questions about inflammatory cells; additional clarity in Figs 3, 4, 8, and 9, and some spelling and grammar corrections.

We look forward to receiving your revised manuscript.

Kind regards,

Rory Edward Morty

Academic Editor

PLOS ONE

Journal Requirements:

3.PLOS ONE now requires that authors provide the original uncropped and unadjusted images underlying all blot or gel results reported in a submission’s figures or Supporting Information files. This policy and the journal’s other requirements for blot/gel reporting and figure preparation are described in detail at https://journals.plos.org/plosone/s/figures#loc-blot-and-gel-reporting-requirements and https://journals.plos.org/plosone/s/figures#loc-preparing-figures-from-image-files. When you submit your revised manuscript, please ensure that your figures adhere fully to these guidelines and provide the original underlying images for all blot or gel data reported in your submission. See the following link for instructions on providing the original image data: https://journals.plos.org/plosone/s/figures#loc-original-images-for-blots-and-gels.

6.In your Methods, please provide full details of animal care and housing.

Reviewers' comments:

Reviewer's Responses to Questions

**Comments to the Author**

1. Is the manuscript technically sound, and do the data support the conclusions?

Reviewer #1: Yes

Reviewer #2: Yes

2. Has the statistical analysis been performed appropriately and rigorously? 

Reviewer #1: Yes

Reviewer #2: Yes

3. Have the authors made all data underlying the findings in their manuscript fully available?

Reviewer #1: Yes

Reviewer #2: Yes

4. Is the manuscript presented in an intelligible fashion and written in standard English?

Reviewer #1: Yes

Reviewer #2: Yes

5. Review Comments to the Author

Reviewer #1: In an ovine model of LPS-induced chorioamnionitis the study aims to determine the therapeutic effect of the IL-1 receptor inhibitor rytvela. The efficacy is assessed by determination of inflammatory responses using a variety of techniques. Methods and statistics are sound and the conclusions are drawn from the results. Rytvela showed a rather small inhibition of inflammation, but as a pilot study this justifies further research and makes the study interesting to readers. Some minor issues should be addressed:

Please cite the literature that the used rytvela concentration is based on.

Please explain the lack of an inflammatory response to LPS concerning TNF-a, IL-8 and IL-1b levels.

As Fig. 3 displays a pre-intervention analysis, no results for rytvela are shown, so please remove rytvela from the text and figure, as this is confusing.

Line 258: please correct Mann-Whitney test

In the discussion the advantages of rytvela compared to other IL-1b antagonists should be discussed, as this is an important aspect of consideration for a future clinical translation. Ideally rytvela should be directly compared to a common IL-1b antagonist, but the reviewer understands that this is out of scope for the current study. A comment in the discussion would be appreciated.

Reviewer #2: Takahashi et al present the results of intra-amniotic administration of non-competitive interleukin-1 receptor antagonist rytvela in an extremely preterm sheep model of chorioamnionitis. The study provides preliminary pre-clinical evidence for the use of Rytvela in prevention of PTB. The manuscript may be improved with the consideration of the following:

- What was the rationale of the dose of LPS used? Can the authors cite prior publications that have used a similar dose?

- Please include the number of biological replicates in all figure legends and represent the individual samples on the plots.

- What is the clinical relevance of inflammation on the skin surface as this is a main finding of the manuscript?

- In Figure 4, TNF-alpha had additional time-points at 36- and 60-hours while the other mediators do not.

- Figure 8 legend also has description for results from Figure 9A.

- Can the authors clarify the relation between IL1-beta and MCP-1 expression? Since the AF did not show any changes in IL-beta but did show a decrease in MCP-1, this is worthy of further clarification. Does it have off-target effects?

- What was the rationale behind the neuronal markers studied?

- Figures 2 and 3 can be combined

- Was any staining used specifically for the identification of the inflammatory cells?

6. PLOS authors have the option to publish the peer review history of their article (what does this mean?). If published, this will include your full peer review and any attached files.

Reviewer #1: No

Reviewer #2: No

---

## [Author Response · Author response to Decision Letter 0]

29 Jun 2021

On behalf of the author team I would like to thank the reviewers for their positive assessment of our work. We have responded to queries in point form, as below, and identified changes in the manuscript (please also see track changes in the uploaded manuscript).

Reviewer #1: 

R1C1: Please cite the literature that the used rytvela concentration is based on.

We have included references to the papers from which the dose of rytvela used in this study was derived.

Line 147-148

R1C2: Please explain the lack of an inflammatory response to LPS concerning TNF-a, IL-8 and IL-1b levels.

The Reviewer raises a very important point; the half-life time and dynamics of the cytokines/chemokines are different depending on fetal organs or amniotic fluid. The expression of RNA transcripts for IL-1β in lung, skin and colon and TNF-α in fetal skin and chorioamnion were significantly increased in LPS Group compared with Saline Group. Additionally, histological assessment of fetal lung and chorioamnion in the LPS Group also showed that an inflammatory response had occurred in response to LPS exposure. 

With regards the lack of a significant difference in amniotic fluid inflammation, it is possible that the effect of surgery and the production of AF cytokines over the 48h recovery period increased the experimental noise to a level sufficient to mask the LPS-driven inflammatory response. We have included reference to these potential factors in the revised manuscript.

Line 456-457, 472-474

R1C3: As Fig. 3 displays a pre-intervention analysis, no results for rytvela are shown, so please remove rytvela from the text and figure, as this is confusing.

Apologies for not clearly articulating the Figure 3. We have changed “the LPS Group and the LPS + rytvela Group” to “LPS-treated Groups”. We have combined Figure 2 and Figure 3 as the Reviewer#2 mentioned. Therefore Figure 2-B is the now previous Figure 3.Line 284-286, 288-290, 299, 302-304 and Figure 2-A, Figure 2-B

R1C4: Line 258: please correct Mann-Whitney test

Thank you for pointing this out – error now corrected.

Line 259

R1C5: In the discussion the advantages of rytvela compared to other IL-1b antagonists should be discussed, as this is an important aspect of consideration for a future clinical translation. Ideally rytvela should be directly compared to a common IL-1b antagonist, but the reviewer understands that this is out of scope for the current study. A comment in the discussion would be appreciated.

We agree that this is a useful addition to the study. Unlike other IL-1β antagonists, rytvela exerts functional selectivity and doesn’t inhibit NF-κB activation. Considering that previous studies have suggested complete inhibition of NF-κB activation may be undesirable for pregnancy, this potential advantage of rytvela has been highlighted as it might be an important feature for any future clinical translation.

Line511-516

Reviewer #2: 

R2C1: What was the rationale of the dose of LPS used? Can the authors cite prior publications that have used a similar dose?

We have previously used a 10 mg dose LPS to the amniotic cavity in the preterm sheep model, which reliably induces chorioamnionitis and a fetal inflammatory response. We have reference to several previous publications using this LPS dose.

Line90-92

R2C2: Please include the number of biological replicates in all figure legends and represent the individual samples on the plots.

We have amended figure legends as suggested.

R2C3: What is the clinical relevance of inflammation on the skin surface as this is a main finding of the manuscript?

The fetal skin is the one of the first tissues to be exposed to microbial stimulation following infection/colonization of the amniotic fluid into fetal body. Although the skin inflammation may not cause localized damage (as is seen with lung or skin) it may have the ability to generate a systemic inflammatory signal given its large percentage of total fetal weight and extensive vascular supply. As this study was designed to explore the presence of fetal tissue inflammation in the setting of LPS and rytvela treatment is important asses fetal skin inflammation. We have updated the revised manuscript accordingly to better reflect this.

Line491-493

R2C4: In Figure 4, TNF-alpha had additional time-points at 36- and 60-hours while the other mediators do not.

Thank you for pointing this out. We have standardized the time points between figures. 

Figure 3

R2C5: Figure 8 legend also has description for results from Figure 9A.

Line 389-392 is part of the main manuscript, and is not part of the legend for Figure 8. Apologies if this was not clear. 

R2C6: Can the authors clarify the relation between IL1-beta and MCP-1 expression? Since the AF did not show any changes in IL-beta but did show a decrease in MCP-1, this is worthy of further clarification. Does it have off-target effects?

MCP-1 is a downstream product of IL-1 β signaling. rytvela selectively inhibits IL-1Racp. Thus, because rytvela doesn’t inhibit or extinguish IL-1β signaling directly, the significant decreases of MCP-1 in AF of the LPS + rytvela Group suggest that rytvela is acting to inhibit, at least in part, downstream of the IL-1 activation. 

Considering the IL-1β concentrations of fetal lung, LPS animals showed significant increases compared with Saline Group, while there was no differences between the LPS + rytvela Group and the Saline Group. IL-1β mRNA expression in the skin of fetuses in the LPS + rytvela Group was significantly lower than that of animals from the LPS Group. As such, our data show that rytvela reduced MCP-1 in the fetal skin but not in the amniotic fluid. 

Taken together, the significant decreases of MCP-1 expression in AF suggested a partial inhibition of inflammation induced by rytvela rather than off-target effects. 

As for the results of the post-intervention analysis (Figure 3), we believe that the combined effects of a limited number of animals and time-course samples combined with experimental noise post-surgery impacted our ability to identify potential differences in amniotic fluid cytokine levels. In the future, it may be worthwhile to avoid a 48h recovery period in inflammatory studies so as to minimize the amount of surgery-induced background inflammation.

Line 541-543

R2C7: What was the rationale behind the neuronal markers studied?

A fetal inflammatory response mediated by cytokine/chemokine expression leads to brain damage in both human fetus, and in animal inflammation models when exposed to LPS. Reactions of the fetal brain to LPS inflammation include: 1) activated microglia, 2) activated astrocytes, 3) maturation arrest or death of oligodendrocytes, and 4) axonal loss or death of neurons. Therefore we chose to assess cellular changes in the brain using the following markers; GFAP for astrocytes, IBA-1 for microglia, Olig-2 for oligodendrocytes and NeuN for neurons, respectively. 

Line392-394

R2C8: Figures 2 and 3 can be combined

We agree and have amended accordingly.

Figure 2-A, B

R2C9: Was any staining used specifically for the identification of the inflammatory cells? 

No, we didn’t conduct other staining to look for specific cell populations. This will be an important part of future studies using this agent. 

N/A

---

## [Decision Letter · Decision Letter 1]

13 Sep 2021

Direct administration of the non-competitive interleukin-1 receptor antagonist rytvela transiently reduced intrauterine inflammation in an extremely preterm sheep model of chorioamnionitis

PONE-D-21-12998R1

Dear Dr. Kemp,

We’re pleased to inform you that your manuscript has been judged scientifically suitable for publication and will be formally accepted for publication once it meets all outstanding technical requirements.

Kind regards,

Kang Sun

Academic Editor

PLOS ONE

Additional Editor Comments (optional):

Reviewers' comments:

Reviewer's Responses to Questions

**Comments to the Author**

1. If the authors have adequately addressed your comments raised in a previous round of review and you feel that this manuscript is now acceptable for publication, you may indicate that here to bypass the “Comments to the Author” section, enter your conflict of interest statement in the “Confidential to Editor” section, and submit your "Accept" recommendation.

Reviewer #1: All comments have been addressed

Reviewer #3: All comments have been addressed

2. Is the manuscript technically sound, and do the data support the conclusions?

Reviewer #1: Yes

Reviewer #3: Yes

3. Has the statistical analysis been performed appropriately and rigorously? 

Reviewer #1: Yes

Reviewer #3: Yes

4. Have the authors made all data underlying the findings in their manuscript fully available?

Reviewer #1: Yes

Reviewer #3: Yes

5. Is the manuscript presented in an intelligible fashion and written in standard English?

Reviewer #1: Yes

Reviewer #3: Yes

6. Review Comments to the Author

Reviewer #1: (No Response)

Reviewer #3: The authors has addressed all the comments raised by the reviewers, and the manuscript is improved by the revisions.

Mnior revision: The number in the manuscript needs to be unified, 3) should be changed to iii) in lines 40 and 143.

7. PLOS authors have the option to publish the peer review history of their article (what does this mean?). If published, this will include your full peer review and any attached files.

Reviewer #1: No

Reviewer #3: No

---

## [Editor Report · Acceptance letter]

17 Sep 2021

PONE-D-21-12998R1 

Direct administration of the non-competitive interleukin-1 receptor antagonist rytvela transiently reduced intrauterine inflammation in an extremely preterm sheep model of chorioamnionitis 

Dear Dr. Kemp:

I'm pleased to inform you that your manuscript has been deemed suitable for publication in PLOS ONE. Congratulations! Your manuscript is now with our production department. 

Kind regards, 

on behalf of

Dr. Kang Sun 

Academic Editor

PLOS ONE